materials science/nanotechnology

energy storage, transition metal dichalcogenide, sodium batteries, supercapacitors, exfoliation

**Author for correspondence:**
Gurpreet Singh
e-mail: gurpreet@ksu.edu

This article has been edited by the Royal Society of Chemistry, including the commissioning, peer review process and editorial aspects up to the point of acceptance.

# Exfoliated transition metal dichalcogenide nanosheets for supercapacitor and sodium ion battery applications

Santanu Mukherjee[1], Jonathan Turnley[1], Elisabeth Mansfield[2], Jason Holm[2], Davi Soares[1], Lamuel David[1] and Gurpreet Singh[1]

[1]Department of Mechanical and Nuclear Engineering, Kansas State University, Manhattan, KS 66506, USA
[2]National Institute of Standards and Technology, Boulder, CO 80305, USA

SM, 0000-0002-1190-1117; EM, 0000-0003-2463-0966; DS, 0000-0002-6369-6565; GS, 0000-0002-1719-9525

Growing concerns regarding the safety, flammability and hazards posed by Li-ion systems have led to research on alternative rechargeable metal-ion electrochemical storage technologies. Among the most notable of these are Na-ion supercapacitors and batteries, motivated, in part, by the similar electrochemistry of Li and Na ions. However, sodium ion batteries (SIBs) come with their own set of issues, especially the large size of the $Na^+$ ion, its relatively sluggish kinetics and low energy densities. This makes the development of novel materials and appropriate electrode architecture of absolute significance. Transition metal dichalcogenides (TMDs) have attracted a lot of attention in this regard due to their relative ease of exfoliation, diverse morphologies and architectures with superior electronic properties. Here, we study the electrochemical performance of Mo-based two-dimensional (2D) layered TMDs (e.g. $MoS_2$, $MoSe_2$ and $MoTe_2$), exfoliated in a superacid, for battery and supercapacitor applications. The exfoliated TMD flakes were interfaced with reduced graphene oxide (rGO) to be used as composite electrodes. Electron microscopy, elemental mapping and Raman spectra were used to analyse the exfoliated material and confirm the formation of 2D TMD/rGO layer morphology. For supercapacitor applications in aqueous electrolyte, the sulfide-based TMD ($MoS_2$) exhibited the best performance, providing an areal capacitance of 60.25 mF cm$^{-2}$. For SIB applications, TMD electrodes exhibited significantly

higher charge capacities than the neat rGO electrode. The initial desodiation capacities for the composite electrodes are 468.84 mAh g$^{-1}$ (1687.82 C g$^{-1}$), 399.10 mAh g$^{-1}$ (1436.76 C g$^{-1}$) and 387.36 mAh g$^{-1}$ (1394.49 C g$^{-1}$) for $MoS_2$, $MoSe_2$ and $MoTe_2$, respectively. Also, the $MoS_2$ and $MoSe_2$ composite electrodes provided a coulombic efficiency of near 100 % after a few initial cycles.

# 1. Introduction

Batteries and supercapacitors are being considered for powering an increasingly diverse range of applications from stationary solar/wind energy farms, to electronic devices like wearable electronics and microchips [1]. To date, lithium-ion rechargeable systems have been at the forefront of devices for energy storage applications [2,3]. Advantages of Li-ion systems include the easy and reversible Li intercalation-deintercalation reaction due to lithium's small ionic size, portability and reasonably high volumetric energy densities [4,5]. However, Li-ion rechargeable systems also have their share of drawbacks, including the high cost of Li metal, flammability, undesirable side reactions and the formation of unwanted solid electrolyte interface (SEI) layers to name a few [6,7]. The high cost of Li especially hinders applications where large quantities are needed, i.e. in medium- and large-scale grid storage applications [8]. These drawbacks have encouraged an ongoing search for non-Li-ion-based rechargeable energy storage devices, among which Na, Mg, Al and K-ion systems are at the forefront [9–11]. Out of these, Na-ion rechargeable systems have received increasing attention because, as opposed to Li, Na resources are practically inexhaustible and evenly distributed around the world [8,12–14]. Also, since Na and Li are both alkali metals, it is likely that electrode materials for Li can be used for Na-ion systems [15]. A simple exchange, however, is not necessarily straightforward because Na and other metal-ions are much larger than Li-ions, and electrode designs must be altered to accommodate: (a) their slow diffusion kinetics in the bulk, and (b) structural changes in the host material associated with large volume changes upon bulky Na-ion insertion/extraction that causes capacity degradation [16,17].

As noted above, the application of Na-ions as charge carriers imposes unique limitations on the electrode material, and therefore the design and development of novel materials and their architectures are of absolute importance [18]. Transition metal dichalcogenides (TMDs) have drawn significant research attention lately because of their unique properties [19–21]. This class of materials can offer two-dimensional (2D) layered morphologies that exhibit large surface areas, and enhanced electrochemical kinetics coupled with low volume changes upon ionic intercalation. For example, Balasingam *et al.* studied few-layered $MoSe_2$ nanosheets prepared by hydrothermal techniques for supercapacitor applications [20]. An optimum capacitance of 49.7 F g$^{-1}$ was obtained with a 75% capacitance retention after 10 000 cycles of operation. $WS_2$ 'nanoribbons' have been studied as a supercapacitor electrode in a potassium electrolyte, and results have indicated the 1 T atomic configuration to perform better than its 2H counterpart [22]. The 1 T nanoribbon produced an optimum areal capacitance of 2813 µF cm$^{-2}$ which was about 12 times greater than the 2H type [22]. A recent report on $MoS_2$ electrodes for Na-ion batteries showed double the capacity value of the Na/$MoS_2$ theoretical capacity [23]. This observation is important because graphite, which can also exhibit 2D layered morphology, exhibits almost negligible capacity and cyclability towards Na [24]. It was speculated that a conversion-type reaction could be responsible for high capacity. $MoSe_2$ nanoplates have been studied as anodes in sodium ion batteries (SIBs). The nanoplates were fabricated by pyrolysis and initial charge and discharge capacities of 1400 C g$^{-1}$ and 1846.8 C g$^{-1}$ were obtained at a rate of 0.1 C. However, the capacity decayed to about 1400 C g$^{-1}$ after 50 cycles of operation. The reduction was attributed to change in the lattice structure of $MoSe_2$ from octahedron to a tetrahedral due to Na ion intercalation [25]. David *et al.* have studied $MoS_2$/graphene composite free-standing anodes in SIB systems [23]. Their composite anode demonstrated a stable charge capacity of 828 C g$^{-1}$ with a 99% coulombic efficiency.

A detailed comparative study of the family of TMDs, especially as electrodes in sodium ion supercapacitor systems, is lacking in the literature. In this manuscript, a novel superacid-assisted exfoliation technique is used to study the electrochemical behaviour of TMDs as potential electrode materials for sodium ion supercapacitors and batteries. This manuscript, therefore, aims to study the effect of exfoliating these materials in a novel way, as well as to study their viability as energy storage materials and to probe into their electrochemical phenomena under both aqueous and organic electrolyte environments.

# 2. Experimental

## 2.1. Synthesis of exfoliated or acid-treated TMD nanosheets

TMD powders (99%, Sigma Aldrich)[1] were sonicated for 30 min in concentrated superacid (pH < −1 and density 1.75 g ml$^{-1}$, Sigma™) to form a solution (concentration of solution ≈ 2 mg ml$^{-1}$) for exfoliating TMD flakes. Two types of superacids were used: chlorosulfonic acid (superacid, 99%, Sigma Aldrich) and methanesulfonic acid. Note that the superacid was added *very slowly* (so as to prevent an exothermic reaction) to the TMD powder in an argon-filled glovebox (dew point −50°C, the low dew point being necessary so that the moisture content is minimal as superacid is to be used). The solution was then carefully quenched in 1.0 l of distilled water (performed with extreme caution in a glovebox). Additional dilution with DI water was done to reduce the solution acidity. The supernatant containing the (lighter) exfoliated flakes was pipetted from the top portion of the solution and dried in a conventional oven to obtain dry superacid treated TMD nanosheets. This process is based on the rationale that the protonation of the TMDs in solution (by the dissociation of the chlorosulfonic acid), coupled with the sonication, provides the energy necessary to overcome the weak van der Waals forces of attraction between the bulk TMD sheets.

## 2.2. Synthesis of graphene oxide

Sodium nitrate, potassium permanganate, sulfuric acid, hydrogen peroxide (31.3% solution in water), hydrochloric acid (30% solution in water), and methanol (99.9%) were purchased from Fisher Scientific. All materials were used as received without further purification. Hummer's method was used to make graphene oxide [26].

## 2.3. Synthesis of rGO/TMD composite paper

A total of 15 mg of GO and 22.5 mg of acid-treated TMD nanosheets were added to a 1 : 1 volume fraction solution of water and isopropanol. The mixture was sonicated for 60 min (Branson Sonifier) to disperse the nanosheets in the solution. The composite suspension was then vacuum-filtered using a 47 mm diameter, 10 µm pore size, filter membrane (HPLC grade, Millipore). TMD/GO composite paper thus obtained was dried in an oven at 70°C overnight and subsequently reduced at 500°C for 2 h, and again at 900°C for 5 min in argon. The samples are denoted hereafter as rGO for neat rGO, and $MoS_2$, $MoSe_2$, and $MoTe_2$ for the respective TMDs, each of which comprises approximately 20% by mass rGO as the conducting agent.

## 2.4. Instrumentation for structural characterization

Transmission electron microscope (TEM) images were digitally acquired by the use of a Phillips CM100 operated at 100 kV. Raman spectroscopy was performed by a Horiba Jobin Yvon LabRam ARAMIS confocal Raman microscope using a He-Ne laser (approx. 632.8 nm). Energy dispersive spectra (EDS) were collected using a Zeiss Gemini scanning electron microscope (SEM) at 10–30 keV. To observe the layered electrode morphology cross-section, a focused $Ga^+$ ion beam (Zeiss Auriga FIB-SEM) was used to cut notches at the edges of the exfoliated TMD composite structures.

## 2.5. Fabrication of electrodes, cell assembly and electrochemical testing

For use in *supercapacitors in aqueous electrolyte*, 20–30 mg of active material (TMD powder) was thoroughly ground with 5% by mass PVDF (binder) and C black (for improving electronic conductivity) each. The grinding was done in a mortar-pestle with the dropwise addition of *N*-methyl pyrrolidone (NMP) solvent such that a viscous slurry was obtained with a 'honey-like' consistency. The slurry was then pasted on stainless steel substrates and dried overnight in a box-furnace at 55°C. For electrochemical testing as supercapacitors, a 3-electrode set-up was used with a Pt wire counter-electrode, and an Ag/AgCl reference electrode in a 1 M aqueous $Na_2SO_4$ electrolyte. Cyclic voltammetry (CV) was performed within a range of 0.0–0.8 V at scan rates of 5, 50, 100, 200 and 500 mV s$^{-1}$, using a CHI™

[1]Certain commercial equipment, instruments, or materials are identified in this paper in order to specify the experimental procedure adequately. Such identification is not intended to imply recommendation or endorsement by the National Institute of Standards and Technology, nor is it intended to imply that the materials or equipment identified are necessarily the best available for the purpose.

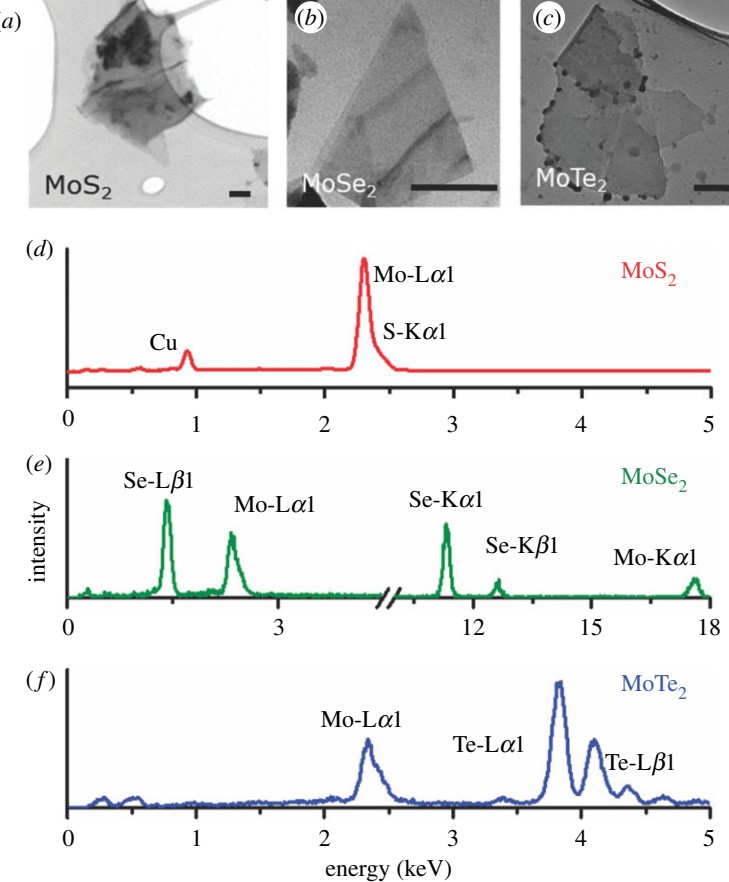

**Figure 1.** TEM micrographs and EDS of the exfoliated TMD samples. Panels (*a*–*c*) are TEM images of the exfoliated TMD samples exhibiting a homogeneous multi-layered nanosheet morphology. Scale bars = 100 nm, (*d*) EDS of the $MoS_2$ sample demonstrating the characteristic Mo and S peaks, along with a Cu peak from the substrate (*e*) EDS of the $MoSe_2$ sample demonstrating the characteristic Mo and Se peaks, (*f*) EDS of the $MoTe_2$ sample indicating characteristic Mo and Te peaks.

660 E electrochemical workstation. This same workstation was used for performing electrochemical impedance spectroscopy (EIS) in 1 M aqueous $Na_2SO_4$ solution. Galvanostatic charge-discharge (GCD) was performed at current densities of 0.5, 1.0, 1.5 and 2.0 mA cm$^{-2}$ using the same workstation. The set-up of the 3-electrode system is such that the electrode needs to be supported by a clip and a free-standing electrode's structural integrity was compromised under the tensile stress from the clip. Hence, a stainless steel current collector was used during this testing.

For applications in *batteries in organic electrolyte*, the free-standing rGO/TMD composite papers were used directly as the working electrodes. 14.3 mm diameter electrodes were punched out of the composite paper. Electrochemical analysis was performed using coin cells (CR 2032) and a two-electrode set-up was employed. 1 M $NaClO_4$ (Alfa Aesar) in (1 : 1 volume fraction) dimethyl carbonate : ethylene carbonate (DMC:EC) served as the electrolyte. A 25 µm thick glass separator soaked in electrolyte was placed between the working electrode and pure Na metal (14.3 mm diameter, 75 µm thick) to be used as the counter-electrode. GCD analysis was performed between 2.25 V and 10 mV versus Na/Na$^+$ using a multichannel BT2000 Arbin test unit.

# 3. Results and discussion

## 3.1. Structural and phase characterization

### 3.1.1. TEM imaging and EDS

TEM imaging is necessary not only to understand the structure and morphology of the exfoliated sheets, but also to observe whether the acid-based exfoliation process has been successful in providing the sheet-like microstructure that is desired. The TEM images of the exfoliated TMD samples in figure 1*a–c*

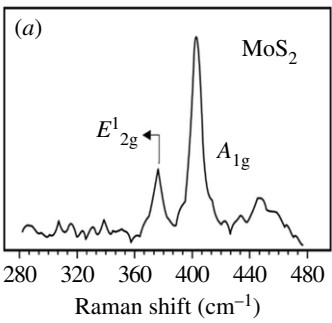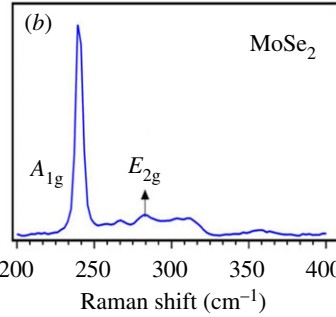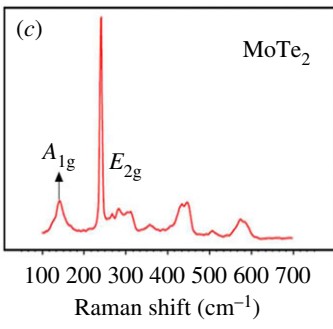

**Figure 2.** Raman spectra of the exfoliated TMD samples. Panels (a–c) are the Raman spectra of the exfoliated $MoS_2$, $MoSe_2$ and $MoTe_2$ samples. The characteristic vibration modes of the sample ($A_{1g}$ and $E_{2g}$) corresponding to their out-of-plane and in-plane vibrations, seen as peaks in the spectra have been labelled for the respective TMDs.

demonstrate the formation of triangular or rhomboidal sheets. The images indicate the formation of a stacked, few-layered morphology and the absence of single-layered flakes can be attributed to the volume/morphology of the bulk TMD precursor, the type and strength of acid used, and the power used during the sonication process [27]. Optimization of the sonication process should be able to provide a direct correlation with number of layers and the type of morphology obtained. Also, the observation that the flakes range in linear dimensions between 50 and 100 nm is consistent with earlier work by the same group [23]. The exfoliated TMD sample images demonstrate a homogeneous and defect-free morphology, especially at the sheet edges. It can be inferred from these observations that the homogeneous nature of the flaky and rigid nanosheets will tend to prevent their restacking and maintain their 2D morphology throughout the cell cycling. It is also pertinent to note that in figure 1c, some dots are observed at the edge of the $MoTe_2$ sample. These dots seen at the edge of the $MoTe_2$ can have two probabilities. Firstly, it can be hypothesized that very small amounts of the exfoliated $MoTe_2$ may have restacked together. This is a feature of exfoliated TMDs and a similar appearance has also been reported in the literature [28,29]. Secondly, it is also possible that all of the bulk $MoTe_2$ obtained commercially may not have been in layered form and there may have been a very small fraction in a non-layered form. Consequently, on exfoliation, these non-layered parts did not exfoliate out and are observed as particulate 'dots' in the TEM. Of course, they are a very small fraction of the entire sample and the TEM image proves that.

Figure 1d shows an EDS of the exfoliated $MoS_2$ sample where characteristic Mo and S peaks (L$\alpha$ and K$\alpha$) can be observed. Figure 1e,f similarly show EDS survey spectra of the exfoliated $MoSe_2$ and the exfoliated $MoTe_2$ samples. The characteristic Mo (K$\alpha$ and L$\alpha$) and Se (K$\alpha$, K$\beta$ and L$\beta$) and Te (L$\alpha$ and L$\beta$) spectral lines can be observed. Another important observation is the absence of any impurity peaks, especially Cl$^-$ peaks from residual chlorosulfonic acid. This indicates the phase purity of the exfoliated samples, thereby further confirming the results seen in the Raman spectra for these samples. However, it is understood that there may be some trace amounts of Cl$^-$ which is beyond the resolution limit of the EDX instrument.

### 3.1.2. Raman spectra

Figure 2a–c shows Raman spectra of the exfoliated $MoS_2$, $MoSe_2$ and $MoTe_2$, respectively. Characteristic $E_{2g}$ and $A_{1g}$ peaks can be observed for all the samples. The $A_{1g}$ mode indicates the out-of-plane vibration of the chalcogen species, whereas the $E_{2g}$ mode is an indicator of in-plane vibrations of the Mo species [30]. It is to be noted that the observed sharp peaks of the $A_{1g}$ modes are mainly due to the interaction of the incident laser beam with the direct band gap, resulting in resonance [31,32]. The peak positions also indicate the formation of few-layered samples. For a monolayer, there is only one active vibrational mode that can be usually observed. However, for the $MoSe_2$ sample, the $A_{1g}$ mode at $240\ cm^{-1}$ is visible as a sharp peak in figure 2b. Also, the $E1_{2g}$ and the $A_{1g}$ modes are observed in all the cases, indicating the existence of several layers. This is especially true for the $MoTe_2$ sample, which shows several auxiliary peaks indicating the formation of multiple layers [33].

The absence of impurity peaks demonstrates that the superacid-based exfoliation process has not resulted in the dissolution of sulfur or formation of impurity phases. EDS results in the next section support this conclusion.

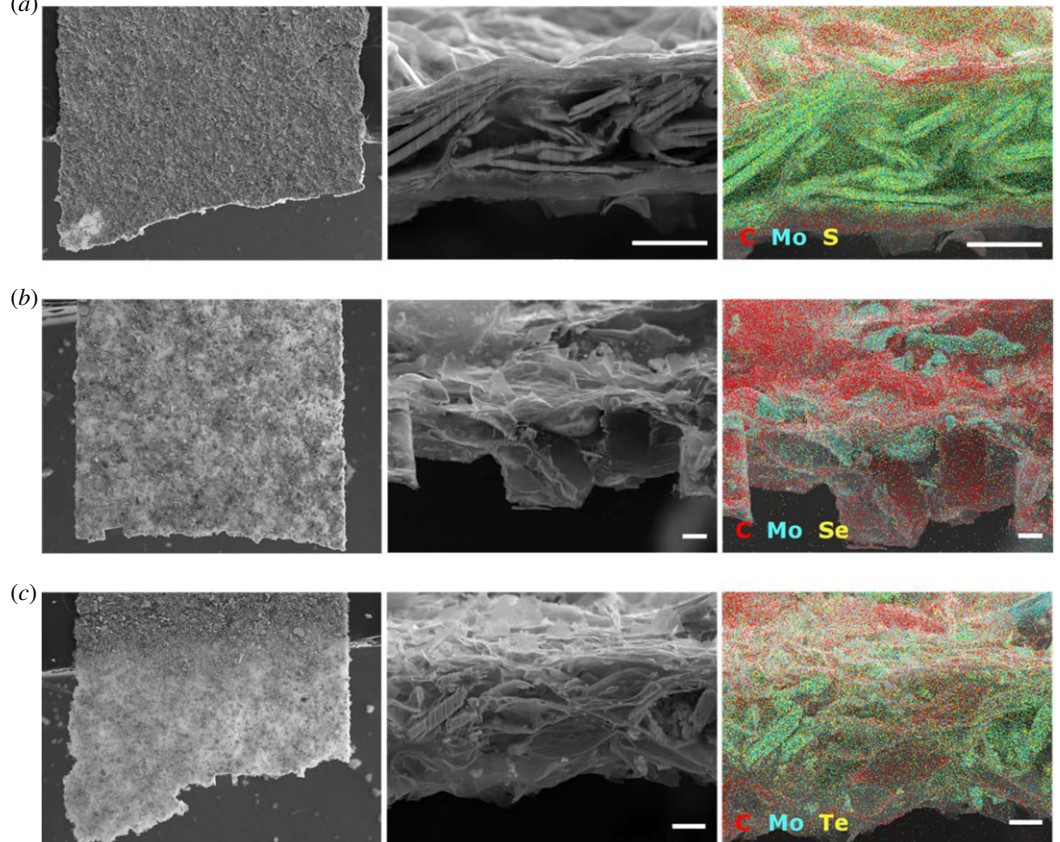

**Figure 3.** Images and elemental maps of the exfoliated, layered structures: (*a*) MoS$_2$, (*b*) MoSe$_2$ and (*c*) MoTe$_2$. Each set of three images shows a low magnification survey image (left, with arrows indicating the 100 nm wide FIB-milled notch), a cross-section view of the structure (middle), and an X-ray map (right) with inset element colour key. Scale bars are 10 μm.

### 3.1.3. Cross-sectional imaging and elemental mapping of the exfoliated layers

Figure 3 shows the 2D exfoliated TMD sample structures in greater detail. The MoS$_2$, MoSe$_2$ and MoTe$_2$ samples are shown in figure 3*a–c*, respectively. Images on the left are low-magnification secondary electron images, in which each sample exhibits a uniform appearance over a large area. The white arrows point to 100 μm wide notches FIB-milled into small pieces of each of the three exfoliated TMD samples. Images in the middle show the notch cross-section where the layered morphology and individual flakes of the TMD material can be observed. Images on the right show X-ray maps superimposed on the secondary electron images. The X-ray maps and the cross-sectional images suggest that the exfoliated MoS$_2$ sample exhibits a distinct layered morphology compared to the MoSe$_2$ and MoTe$_2$ samples, with several TMD flakes sandwiched between rGO layers. It is important to note that the X-ray maps suggest the composition is relatively uniform. If individual 'islands' had been observed throughout the flake, this could have contributed to inhomogeneities that would be detrimental towards electrochemical performance.

## 3.2. Electrochemical performance analysis

### 3.2.1. Supercapacitor performance

Electrochemical performance of the exfoliated TMD powders as supercapacitors is shown in figure 4*a–d*.

Figure 4*a* shows the CV results from exfoliated TMD electrodes collected at a 200 mV s$^{-1}$ scan rate. All samples exhibit a rectangular CV curve, indicating double layer capacitive behaviour and suggesting that charge is stored along the electrode surface. It is observed that greater electrochemical surface areas are obtained as a consequence of the layered morphology, which enhances the ability to store charge at the interface. Increasing scan rates produced larger areas enclosed by the CV curve along with higher peak currents; however, no distinct redox peaks were observed. Figure 4*b* shows the capacitances obtained for

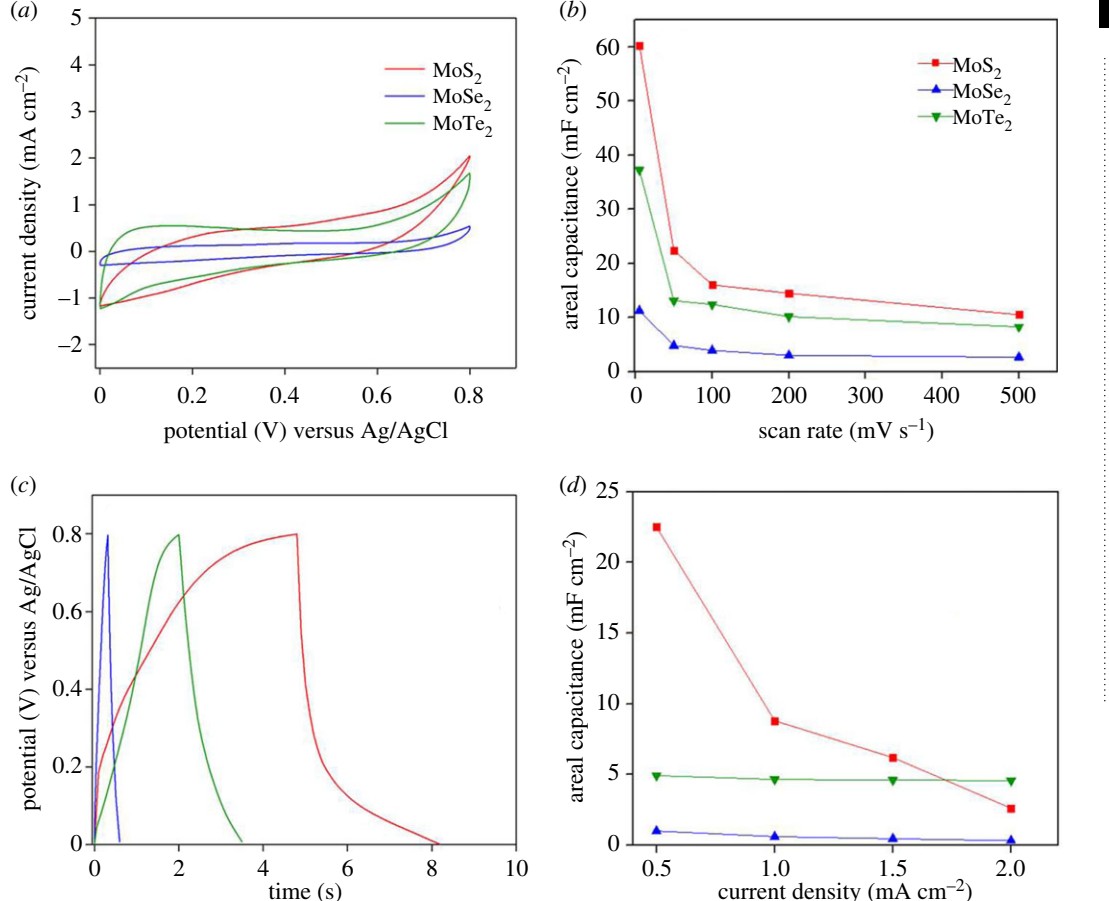

**Figure 4.** Electrochemical performance of exfoliated TMD electrodes in aqueous media as supercapacitor electrodes. (*a*) CV of the exfoliated TMD electrodes in 1 M aq. Na$_2$SO$_4$ performed at a scan rate of 200 mV s$^{-1}$, showing characteristic double layer capacitance behaviour, (*b*) corresponding specific capacitances of the TMD electrodes at 5, 50, 100, 200 and 500 mV s$^{-1}$ with the MoS$_2$ electrode providing the best performance, (*c*) first cycle GCD curves of the exfoliated TMD electrodes obtained at a current density of 1 mA cm$^{-2}$, (*d*) corresponding specific capacitances for the various TMD electrodes at current densities of 0.5, 1.0, 1.5 and 2.0 mA cm$^{-2}$, with MoS$_2$ demonstrating superior performance.

the same samples at CV scan rates from 5 to 500 mV s$^{-1}$. Results demonstrate that MoS$_2$ performs the best, providing a maximum areal capacitance of 60.25 mF cm$^{-2}$.

Figure 4*c* shows the first cycle GCD curves of the exfoliated TMD electrodes. The symmetric GCD curves correspond to a reversible electrochemical process with reduced hysteresis losses [34]. Here too, the MoS$_2$ sample performs better than the others, with a maximum areal capacitance of 22.5 mF cm$^{-2}$ at a current density of 0.5 mA cm$^{-2}$. This can be attributed to the quality of exfoliation leading to better electrochemical properties of MoS$_2$, which fits well with the cross-sectional images seen in figure 3. Figure 4*d* shows that the areal capacitances converge to similar values for all the samples at high current rates (approx. 2 mA cm$^{-2}$). This can be attributed to the significantly reduced ionic diffusion at such high current rates. These results corroborate ab initio studies of MoSe$_2$ electrodes as supercapacitors for Na-ion systems [35].

The capacitance values obtained in this manuscript represent the lower limit for the exfoliated TMD samples. This is because the samples were directly used after the exfoliation process without optimizing for the quality of output, i.e. quality and number of layers formed. Also, though characterization techniques demonstrate no impurity phases present, there may be electrochemically competing phases present which may not have been detected. A post-exfoliation 'cleansing' of the TMD samples may also improve the capacitance.

### 3.2.2. Electrochemical impedance spectroscopy

EIS was performed for further understanding the electrochemical behaviour of the materials synthesized. From figure 5, the Nyquist plot shows that for the three TMD/rGO materials studied, the semicircle, at

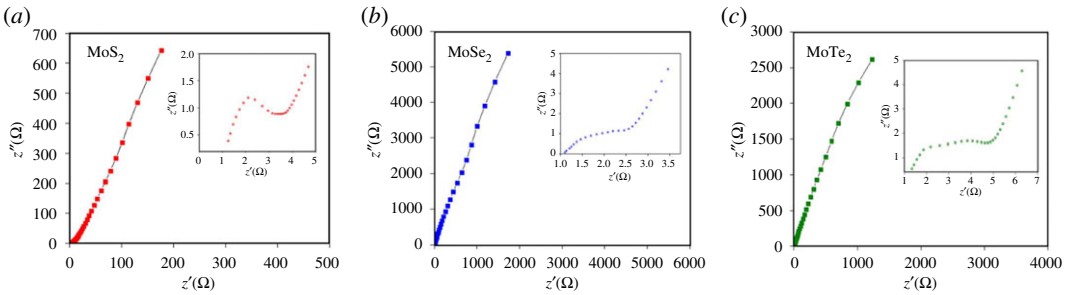

**Figure 5.** Electrochemical impedance spectra of the exfoliated TMD/rGO composite anodes. Panels ($a$–$c$) represent the EIS of the MoS$_2$/rGO, MoSe$_2$/rGO and MoTe$_2$/rGO samples, respectively, performed in aqueous 1 M Na$_2$SO$_4$ solution. All samples exhibit the characteristic inverted semicircle in the high-frequency regime and a linear trajectory in the low-frequency regime.

**Scheme 1** . Equivalent circuit corresponding to the EIS analysis, consisting of the individual resistive and capacitive components.

**Table 1.** Listing of values of the individual resistive and capacitive components for each exfoliated TMD/rGO composite.

| impedance element | MoS$_2$ | MoSe$_2$ | MoTe$_2$ |
| --- | --- | --- | --- |
| $R_{esr}$ ($\Omega$) | 0.905 | 1.20 | 0.684 |
| $R_{ct}$ ($\Omega$) | 4.15 | 3.86 | 5.32 |
| $C_{PE\ EDL} - Y_{EDL}$ ($\mu\Omega^{-1} * s^n$) | 32.9 ($n = 0.702$) | 14.3 ($n = 0.891$) | 31.1 ($n = 0.67$) |
| $C_{PE\ \theta} - Y_\theta$ ($\mu\Omega^{-1} * s^n$) | 387 ($n = 0.763$) | 22.2 ($n = 0.819$) | 61.5 ($n = 0.80$) |

the high-frequency region, is attributed to pseudocapacitive and double-layer (EDL) processes. Importantly, one thing that may contribute toward the fast-charge faradaic processes for TMDs is the fact that such species possess different oxidation states [36]. Thus, the capacitance may be a combination of EDL and faradaic charge storage processes [37].

The canonical circuit, proposed by Tilak *et al.*, was employed to simulate the results [38]. This circuit, as shown in scheme 1, is composed of the following elements: bulk equivalent series resistance of the solution ($R_{esr}$), charge transfer resistance ($R_{ct}$), capacitance of the electric double layer ($C_{PE\ EDL}$), and the capacitance related to the pseudocapacitive reactions ($C_{PE\theta}$) [39]. It is worth mentioning that the capacitors in parallel ($C_{PE\ EDL}$ and $C_{PE\theta}$) are equivalent to $Z_{CPE} = 1/Y_0(j\omega)^n$, a variable which accounts for the dispersion phenomena inherent from electrodes surface, which are non-homogeneous; and for the adsorption of ions from the electrolyte. Thus, these processes are simulated and quantified within the range $0 \leq n \leq 1$.

From the circuit above, it is possible to obtain the contributions by each parameter i.e. bulk equivalent series resistance of the solution ($R_{esr}$), charge transfer resistance ($R_{ct}$), capacitance of the electric double layer ($C_{PE\ EDL}$) and the capacitance related to the pseudocapacitive reactions ($C_{PE\theta}$); these values are provided in table 1.

### 3.2.3. Battery performance

The performance of the TMD composite papers as anodes in Na/Na$^+$ half cells is provided in figure 6$a$–$g$.

Figure 6$a$,$c$,$e$,$g$ are the first and second cycle GCD curves for the exfoliated TMD electrodes (MoS$_2$, MoSe$_2$ and MoTe$_2$) and the rGO electrodes, respectively. The exfoliated MoS$_2$, MoSe$_2$ and the MoTe$_2$ composite paper samples demonstrate the first cycle sodiation (discharge) capacities of 468.84 mAh g$^{-1}$ (1687.82 C g$^{-1}$), 399.1 mAh g$^{-1}$ (1436.76 C g$^{-1}$) and 387.36 mAh g$^{-1}$ (1394.49 C g$^{-1}$),

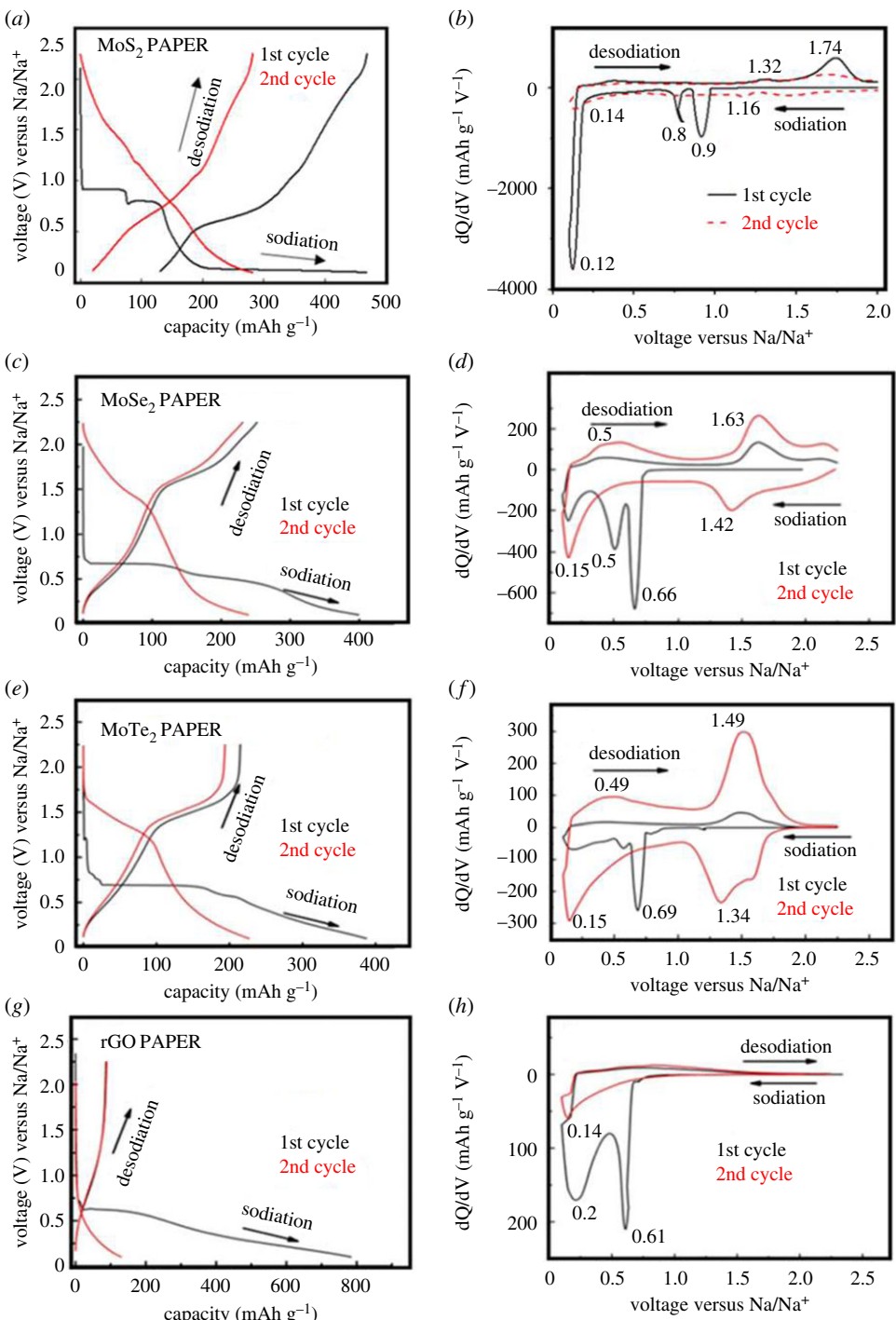

**Figure 6.** Electrochemical performance of TMD composite electrodes in Na/Na$^+$ half-cell. (*a,c,e,g*) are the GCD curves of the first and second cycles for the MoS$_2$, MoSe$_2$, MoTe$_2$ and rGO electrodes demonstrating their characteristic behaviour towards Na$^+$ ion interaction, (*b,d,f,h*) are the corresponding dQ/dV plots of the composite TMD electrodes exhibiting the characteristic sodiation and desodiation peaks. (1 mAh g$^{-1}$ = 3.6 C g$^{-1}$). Permission for reuse of data for figures (*b,g,h*) obtained from ACS publications [23].

whereas their charge (desodiation) capacities are 512.25 mAh g$^{-1}$ (1844.1 C g$^{-1}$), 251.86 mAh g$^{-1}$ (906.73 C g$^{-1}$) and 214.62 mAh g$^{-1}$ (772.66 C g$^{-1}$). These charge capacities are approximately 2.5–3 times greater than that of the rGO electrode, which delivers a first cycle charge capacity of 81.5 mAh g$^{-1}$ (293.4 C g$^{-1}$) as observed in figure 6*h*. The authors would like to point out that these capacities are for free-standing TMD composite paper electrodes without any additive binder contributions. Also, the capacities obtained in the present manuscript are comparatively higher than some of the values that have been reported in the literature [40,41].

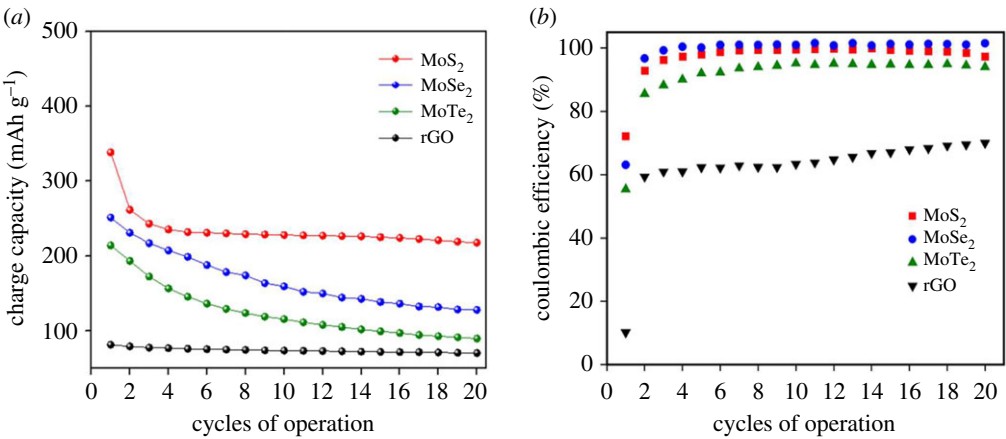

**Figure 7.** Capacity retention and coulombic efficiency curves. (*a*) Capacity retention trend and (*b*) coulombic efficiency trends showing the superior performance of TMD composite electrodes over standalone rGO electrode.

However, the rGO paper showed significantly higher charge capacity than commercial graphite ($293.4 \, \text{C g}^{-1}$ versus approx. $126 \, \text{C g}^{-1}$) for the Na/Na$^+$ half-cell [42]. The voltage profile curves showed that the oxidation/reduction mechanism of TMD composite electrodes with respect to sodium exhibited two distinct plateaus and were very different when compared to the sloped rGO discharge curve. Differential capacity (dQ/dV) curves were plotted (figure 6*b,d,f,h*) to study the Na$^+$ ion interaction with the TMD and the rGO electrodes in greater detail. Since each peak corresponds to a specific reaction and certain peaks disappeared after the first cycle, those peaks (e.g. 0.5 V and 0.6 V for MoSe$_2$, 0.69 V for MoTe$_2$) can be ascribed to the formation of an SEI layer[15]. In the second cycle, all TMD electrodes showed two insertion peaks (approx. 0.15 V and approx. 1.34 V to 1.46 V) and two extraction peaks (approx. 0.5 and approx. 1.5 V to 1.7 V), which indicates distinct sodiation and desodiation in the TMD composite electrodes.

From figure 6*a* MoS$_2$ demonstrates a two-step interaction with Na ions at approximately 0.9 V and approximately 0.75 V. In this interaction, cations are intercalated into the ordered MoS$_2$ structure, leading to a conversion reaction in which the material degrades into Mo and Na$_2$S, as previously reported [23].

For MoSe$_2$, ionic interaction with the structure results in its transformation from a semiconductor to a metal. Wang *et al.* have demonstrated that when Na$^+$ ions intercalate into MoSe$_2$, equal amounts of electrons also enter to maintain overall electrical neutrality and these electrons insert themselves into the $t_{2g}$ orbitals of Mo [25]. It is this that results in the transition to a metallic stage. Also, it has been shown that Na$^+$ ion intercalation can occur at two different sites in the MoSe$_2$ lattice: at a tetrahedral or an octahedral location. For MoSe$_2$, the plateau at approximately 0.65 V corresponds to the formation of Na$_x$MoSe$_2$, whereas the other plateau at around 0.4 V corresponds to reduction of Mo$^{4+}$ to Mo and the formation of Na$_2$Se [25].

The reaction mechanism is similar for MoTe$_2$, which has a plateau corresponding to the formation of metallic Mo and Na$_2$Te [43]. The conversion reaction in this case is as follows:

$$MoX_2 + 4Na \rightleftharpoons Mo + 2Na_2X \quad (X = \text{chalcogen atm e.g. S, Se, Te etc.})$$

It is noted that all the TMD samples undergo a large first cycle capacity loss (especially MoS$_2$), which is attributed to the SEI layer formation which impedes the charge transfer process initially. Also, structural changes occur during the initial cycles as a result of the transformation process and side reactions with the electrolyte also contribute to the first cycle loss. It is to be also noted that the difference in the cathodic and anodic peak locations for the TMDs with different chalcogen atoms are due to interfacial Na$^+$ ion storage, interaction/storage of Na$^+$ ions in different defective sites of each crystal and its constituent layers [44]. It is understood that the presence of rGO enhances the performance of these exfoliated TMDs. rGO not only increases electronic conductivity, but also helps prevent the restacking of the exfoliated TMD layers by acting as support and relieving the stress produced to the cell cycling [23,45].

The neat rGO electrode only showed a broad plateau during the extraction half, indicating insufficient intercalation.

Capacity retention data for various electrodes up to 20 cycles are presented in figure 7*a*. After the completion of 20 cycles, MoS$_2$ demonstrated a capacity retention of approximately 83% (leaving aside

the first cycle loss) followed by MoSe$_2$ (63.44%) and MoTe$_2$ (53.98%). Furthermore, the TMD electrodes reached near 100% cycling efficiency after the second discharge and remained constant up to the 20th cycle. Therefore, an analysis of the GCD and preliminary cell cycling results demonstrates the promise of our novel exfoliated free-standing TMD materials as negative electrodes for SIB applications. Additionally, as seen in figure 7b, all TMD electrodes showed first cycle efficiency in excess of approximately 50% compared to rGO, which demonstrated only 10.16%.

## 4. Conclusion

A superacid-based technique was used to successfully exfoliate Mo-based layered TMDs. Detailed structural and elemental characterization indicates the formation of phase pure few-layered TMD nanosheets, thereby demonstrating the efficacy of the process. The characterization process has also shown that sheets are usually homogeneous and the acid-based process has not resulted in any dissolution of the material. The exfoliated sheets were then combined with rGO to form composite paper electrodes for electrochemical energy storage applications.

For battery configuration, all the TMD composite electrodes perform better than rGO, providing an initial charge capacity of 468.84 mAh g$^{-1}$, 399.10 mAh g$^{-1}$ and 387.36 mAh g$^{-1}$ for MoS$_2$, MoSe$_2$ and MoTe$_2$, respectively. TMD electrodes also provide steady coulombic efficiencies (close to 100% after a few cycles) and capacity values at least approximately 2.5 to 3 times those of neat rGO electrode. The electrochemical analyses, especially the incremental capacity curves, illustrate the Na$^+$ ion interaction mechanisms in the TMDs, with the transformation reaction regimes clearly distinguished for each electrode. The results offer promise for the use of TMD-based electrodes in electrochemical energy storage systems.

Data accessibility. Data available from the Dryad Digital Repository: https://doi.org/10.5061/dryad.tv2tk65 [46].
Authors' contributions. J.T. and S.M. performed supercapacitor testing. L.D. prepared composite paper electrodes and battery tests. G.S. assisted L.D. with cell assembly. E.M and J.H. performed electron microscopy and elemental analysis/mapping. G.S. conceived the idea, designed the experiments and co-wrote the manuscript with S.M., D.S. performed the impedance spectra experiments. All authors discussed the results and commented on or revised the manuscript.
Competing interest. We have no competing interests.
Funding. This work is supported by National Science Foundation grant no. 1454151.

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
