## [Reviewer comments · Royal Society Open Science]

Review History

RSOS-190437.R0 (Original submission)

Review form: Reviewer 1

Is the manuscript scientifically sound in its present form?

Yes

Are the interpretations and conclusions justified by the results?

Yes

Is the language acceptable?

Yes

Is it clear how to access all supporting data?

Yes

Do you have any ethical concerns with this paper?

No

Have you any concerns about statistical analyses in this paper?

No

Recommendation?

Major revision is needed (please make suggestions in comments)

Comments to the Author(s)

Review of "Exfoliated transition metal dichalcogenide (TMD) nanosheets for supercapacitor and sodium ion battery applications", Manuscript ID: RSOS-190437.

In this manuscript, the authors fabricated the TMD nanosheet papers using a superacid exfoliation approach and testified and compared different TMD in their electrochemical performance compared with pure rGO.

To help improve the scientific depth of their study, I raised several questions/suggestions as below, including corrections, further clarification, as well as more result supplemental.

1. What is the authors' definition or criteria for "superacid" in terms of pH or concentration?
2. The authors mentioned dew point of -50C in the glove box. Is it for the superacid solution? In addition, what would happen when pouring superacid into TMD fast?
3. The authors baked the superacid treated TMD flake solutions. Did they wash the flakes before the next step in order to remove the superacid residues?
4. Please state in context the functions of PVDF, C black and NMP in the preparation of electrode?
5. What are the sheet-edge distributed dots in the TEM image of MoTe₂?
6. In Figure 1, EDS data shows that there are relatively more C in the sample of MoS₂ than the other two. Why?
7. The author explained why in the TEM images there are no single-layered nanosheet. Is it also possible that after dilution, the exfoliated nanosheets tend to cluster due to Van der Waals interaction? This question is also somehow related to the purity of nanosheets (see question 3). For example, the Cl residue could exist in very small amount that the EDX data could not find it due to resolution issue.
8. Page 10 Line 38: there is a typo: "The peak positions and the peak position...".
9. Figure 2: please use arrows to help labeling the peaks. For 2(C), it looks like the sharp peak is the E_{2g} peak, which is different than the other two.
10. Figure 3: please use images with the same scale. Do all the prepared three sheet papers have the same thickness and internal "sandwich" cross section?
11. Figure 3C: in term of the colors, it looks like the distribution of Mo and Te has a little local difference. Is that within the error range of EDX mapping?
12. Figure 4B and 4D: how large the error bar (repeatability)?
13. Figure 4B and 4D: typo in the y-axis: "'aerial" should be "areal"?
14. Figure 5G: if TMD/rGO composite paper performs much better than pure rGO, why not preparing pure TMD paper?
15. The author concluded that MoS₂ performs better in capacitor and battery than the counterparts, however no GCD and dQ/dV curves were shown for MoS₂.

Review form: Reviewer 2**Is the manuscript scientifically sound in its present form?**

Yes

Are the interpretations and conclusions justified by the results?

Yes

Is the language acceptable?

Yes

Is it clear how to access all supporting data?

Not Applicable

Do you have any ethical concerns with this paper?

No

Have you any concerns about statistical analyses in this paper?

No

Recommendation?

Accept with minor revision (please list in comments)

Comments to the Author(s)

This manuscript reports the preparation of a collection of Mo-based 2D layered TMDs through a superacid-based technique. Their electrochemical performance is evaluated for both supercapacitors and sodium ion batteries. The authors identify MoS₂ layered structures exhibited the best performance in both applications. Overall, I believe this work represents an interesting new strategy for preparing layered TMDs to systematically study the electrochemical performance of Mo-based TMDs in supercapacitors and sodium ion batteries. However, there are several concerns to be addressed. I would recommend acceptance of this paper after minor revision.

1. As shown in Figure 1 (c), there are some nanoparticles on the edge of the layered MoTe₂. What are those particles?
2. Stainless steel was used as the current collector for supercapacitor application, while a free-standing electrode configuration was adopted for sodium ion batteries. Why is that? Why is the free-standing electrode not used for supercapacitor characterization?
3. In Figure 4 (c), the charge-discharge curve for MoS₂ is not symmetric, suggesting a low coulombic efficiency. The authors should analyze and explain that. In addition, the electrochemical impedance spectroscopy measurements are suggested to be performed to better illustrate the differences among those TMDs.

Decision letter (RSOS-190437.R0)

30-Apr-2019

Dear Dr Mukherjee:

Title: Exfoliated transition metal dichalcogenide (TMD) nanosheets for supercapacitor and sodium ion battery applications

Manuscript ID: RSOS-190437

The editor assigned to your manuscript has now received comments from reviewers. We would like you to revise your paper in accordance with the referee and Subject Editor suggestions which can be found below (not including confidential reports to the Editor). Please note this decision does not guarantee eventual acceptance.

Please submit your revised paper before 23-May-2019. Please note that the revision deadline will expire at 00.00am on this date. If we do not hear from you within this time then it will be assumed that the paper has been withdrawn. In exceptional circumstances, extensions may be possible if agreed with the Editorial Office in advance. We do not allow multiple rounds of revision so we urge you to make every effort to fully address all of the comments at this stage. If deemed necessary by the Editors, your manuscript will be sent back to one or more of the original reviewers for assessment. If the original reviewers are not available we may invite new reviewers.

Please also include the following statements alongside the other end statements. As we cannot publish your manuscript without these end statements included, if you feel that a given heading is not relevant to your paper, please nevertheless include the heading and explicitly state that it is not relevant to your work.

- Funding statement

Please include a funding section after your main text which lists the source of funding for each author.

RSC Associate Editor:
Comments to the Author:
(There are no comments.)

RSC Subject Editor:
Comments to the Author:
(There are no comments.)

Reviewers' Comments to Author:
Reviewer: 1

Comments to the Author(s)

Review of "Exfoliated transition metal dichalcogenide (TMD) nanosheets for supercapacitor and sodium ion battery applications", Manuscript ID: RSOS-190437.

In this manuscript, the authors fabricated the TMD nanosheet papers using a superacid exfoliation approach and testified and compared different TMD in their electrochemical performance compared with pure rGO.

To help improve the scientific depth of their study, I raised several questions/suggestions as below, including corrections, further clarification, as well as more result supplemental.

1. What is the authors' definition or criteria for "superacid" in terms of pH or concentration?
2. The authors mentioned dew point of -50C in the glove box. Is it for the superacid solution? In addition, what would happen when pouring superacid into TMD fast?
3. The authors baked the superacid treated TMD flake solutions. Did they wash the flakes before the next step in order to remove the superacid residues?
4. Please state in context the functions of PVDF, C black and NMP in the preparation of electrode?
5. What are the sheet-edge distributed dots in the TEM image of MoTe₂?
6. In Figure 1, EDS data shows that there are relatively more C in the sample of MoS₂ than the other two. Why?
7. The author explained why in the TEM images there are no single-layered nanosheet. Is it also possible that after dilution, the exfoliated nanosheets tend to cluster due to Van der Waals interaction? This question is also somehow related to the purity of nanosheets (see question 3). For example, the Cl residue could exist in very small amount that the EDX data could not find it due to resolution issue.
8. Page 10 Line 38: there is a typo: "The peak positions and the peak position...".
9. Figure 2: please use arrows to help labeling the peaks. For 2(C), it looks like the sharp peak is the E_{2g} peak, which is different than the other two.
10. Figure 3: please use images with the same scale. Do all the prepared three sheet papers have the same thickness and internal "sandwich" cross section?
11. Figure 3C: in term of the colors, it looks like the distribution of Mo and Te has a little local difference. Is that within the error range of EDX mapping?
12. Figure 4B and 4D: how large the error bar (repeatability)?
13. Figure 4B and 4D: typo in the y-axis: ""aerial" should be "areal"?
14. Figure 5G: if TMD/rGO composite paper performs much better than pure rGO, why not preparing pure TMD paper?
15. The author concluded that MoS₂ performs better in capacitor and battery than the counterparts, however no GCD and dQ/dV curves were shown for MoS₂.

Reviewer: 2

Comments to the Author(s)

This manuscript reports the preparation of a collection of Mo-based 2D layered TMDs through a superacid-based technique. Their electrochemical performance is evaluated for both supercapacitors and sodium ion batteries. The authors identify MoS₂ layered structures exhibited the best performance in both applications. Overall, I believe this work represents an interesting new strategy for preparing layered TMDs to systematically study the electrochemical performance of Mo-based TMDs in supercapacitors and sodium ion batteries. However, there are several concerns to be addressed. I would recommend acceptance of this paper after minor revision.

1. As shown in Figure 1 (c), there are some nanoparticles on the edge of the layered MoTe₂. What are those particles?
2. Stainless steel was used as the current collector for supercapacitor application, while a free-standing electrode configuration was adopted for sodium ion batteries. Why is that? Why is the free-standing electrode not used for supercapacitor characterization?
3. In Figure 4 (c), the charge-discharge curve for MoS₂ is not symmetric, suggesting a low coulombic efficiency. The authors should analyze and explain that. In addition, the electrochemical impedance spectroscopy measurements are suggested to be performed to better illustrate the differences among those TMDs.

Author's Response to Decision Letter for (RSOS-190437.R0)

See Appendix A.

RSOS-190437.R1 (Revision)

Review form: Reviewer 2

Is the manuscript scientifically sound in its present form?

Yes

Are the interpretations and conclusions justified by the results?

Yes

Is the language acceptable?

Yes

Is it clear how to access all supporting data?

Yes

Do you have any ethical concerns with this paper?

No

Have you any concerns about statistical analyses in this paper?

No

Recommendation?

Accept with minor revision (please list in comments)

Comments to the Author(s)

The authors have addressed most of the concerns. However, there are still several issues to be dealt with before acceptance.

1. It seems that the title for Figure 2 is not appropriate. There is no elemental analysis (mapping) shown in Figure 2. Instead, the elemental analysis is shown in Figure 3.
2. In Figure 4, the areal capacitance was calculated and plotted. I am wondering what the loading amount of the active material (TMDs) is for the supercapacitor test as the loading will affect the areal capacitance.
3. In Figure 5, the title for the x-axis in (b) should be as same as that in (d), (f) and (h).
4. Table 1 should be reorganized.
5. The part for electrochemical impedance spectroscopy is related to supercapacitor performance. I would suggest this part should be placed right after the supercapacitor performance.

Decision letter (RSOS-190437.R1)

12-Jun-2019

Dear Dr Mukherjee:

Title: Exfoliated transition metal dichalcogenide (TMD) nanosheets for supercapacitor and sodium ion battery applications

Manuscript ID: RSOS-190437.R1

Thank you for submitting the above manuscript to Royal Society Open Science. On behalf of the Editors and the Royal Society of Chemistry, I am pleased to inform you that your manuscript will be accepted for publication in Royal Society Open Science subject to minor revision in accordance with the referee suggestions. Please find the reviewers' comments at the end of this email.

The reviewers and handling editors have recommended publication, but also suggest some minor revisions to your manuscript. Therefore, I invite you to respond to the comments and revise your manuscript.

Please also include the following statements alongside the other end statements. As we cannot publish your manuscript without these end statements included, if you feel that a given heading is not relevant to your paper, please nevertheless include the heading and explicitly state that it is not relevant to your work. We have included a screenshot example of the end statements for reference.

- Funding statement

Please include a funding section after your main text which lists the source of funding for each author.

Because the schedule for publication is very tight, it is a condition of publication that you submit the revised version of your manuscript before 21-Jun-2019. Please note that the revision deadline

will expire at 00.00am on this date. If you do not think you will be able to meet this date please let me know immediately.

Best wishes,
Dr Laura Smith
Publishing Editor, Journals

RSC Associate Editor:
Comments to the Author:
(There are no comments.)

RSC Subject Editor:
Comments to the Author:
(There are no comments.)

Reviewer comments to Author:
Reviewer: 2

Comments to the Author(s)

The authors have addressed most of the concerns. However, there are still several issues to be dealt with before acceptance.

1. It seems that the title for Figure 2 is not appropriate. There is no elemental analysis (mapping) shown in Figure 2. Instead, the elemental analysis is shown in Figure 3.
2. In Figure 4, the areal capacitance was calculated and plotted. I am wondering what the loading amount of the active material (TMDs) is for the supercapacitor test as the loading will affect the areal capacitance.
3. In Figure 5, the title for the x-axis in (b) should be as same as that in (d), (f) and (h).
4. Table 1 should be reorganized.
5. The part for electrochemical impedance spectroscopy is related to supercapacitor performance. I would suggest this part should be placed right after the supercapacitor performance.

Author's Response to Decision Letter for (RSOS-190437.R1)

See Appendix B.

RSOS-190437.R2 (Revision)

Review form: Reviewer 2

Is the manuscript scientifically sound in its present form?

Yes

Are the interpretations and conclusions justified by the results?

Yes

Is the language acceptable?

Yes

Do you have any ethical concerns with this paper?

No

Have you any concerns about statistical analyses in this paper?

No

Recommendation?

Accept as is

Comments to the Author(s)

The authors have addressed all the concerns. I would recommend acceptance as it is.

Decision letter (RSOS-190437.R2)

16-Jul-2019

Dear Dr Mukherjee:

Title: Exfoliated transition metal dichalcogenide (TMD) nanosheets for supercapacitor and sodium ion battery applications

Manuscript ID: RSOS-190437.R2

It is a pleasure to accept your manuscript in its current form for publication in Royal Society Open Science. The chemistry content of Royal Society Open Science is published in collaboration with the Royal Society of Chemistry.

RSC Associate Editor: 1
Comments to the Author:
(There are no comments.)

RSC Associate Editor: 2
Comments to the Author:
(There are no comments.)

Reviewer(s)' Comments to Author:
Reviewer: 2

Comments to the Author(s)
The authors have addressed all the concerns. I would recommend acceptance as it is.

Appendix A

May 26, 2019

RE: Submission of revised manuscript (**Manuscript id: RSOS 190437**) titled, “*Exfoliated transition metal dichalcogenide (TMD) nanosheets for supercapacitor and sodium ion battery applications*”.

Dear Editors:

We are submitting the revised manuscript (**Manuscript id: RSOS 190437**) titled, “*Exfoliated transition metal dichalcogenide (TMD) nanosheets for supercapacitor and sodium ion battery applications*”. Our submission will include the following:

1. This cover letter along with the following Appendix:
 - **Appendix I** which shows our response to the editorial comments in a point-by-point fashion, attached with this cover letter.
2. **Revised primary manuscript file** (without any highlights and markups), which has all the updates and revisions we have made.

We will also be uploading a **Supplementary material** (revised manuscript with a markups showing the revisions and changes made)

We would also like to add that we have added a new co-author, Mr. Davy Soares, a current PhD student in the group who performed the impedance spectra experiments for the revised manuscript. His contribution has also been updated in the appropriate section.

We would like to thank the editors for appreciating our manuscript and taking their time to provide valuable suggestions for further refining our manuscript. Look forward to hearing from you.

Sincerely,

Gurpreet Singh

Harold O. and Jane C. Massey Neff Professor,

Associate Professor, Mechanical and Nuclear Engineering Department

3002 Rathbone Hall, Kansas State University, Manhattan,

Kansas 66506, USA

<http://www-personal.ksu.edu/~gurpreet/>

Phone: 785-532-7085

Appendix I

Reviewer 1.

Reviewer comment 1. What is the authors' definition or criteria for "superacid" in terms of pH or concentration?

Author's response: The authors wish to thank the reviewer for this valuable question. The concentration of the chlorosulfonic acid considered was 99 %, a density of 1.75 g ml⁻¹ at room temperature and pH < -1, obtained from Sigma Aldrich™.

Changes made: The information regarding the pH and density of the superacid have been incorporated in the revised manuscript, in page 5, lines 3 and 4).

Reviewer comment 2. The authors mentioned dew point of -50 C in the glove box. Is it for the superacid solution? In addition, what would happen when pouring superacid into TMD fast?

Author's response: The authors wish to thank the reviewer for this question. The low dew point (-50 C) has been specifically set so that the moisture content of the air is minimal inside the glove box. Moisture in the air not only will cause damage to the glove box, a low dew point is absolutely essential here as a superacid is being used inside the glovebox.

A fast addition of superacid to the TMDs may result in a strongly exothermic reaction, fumes being generated and a thermal runoff. Therefore, to prevent this thermal hazard, the chlorosulfonic acid is added dropwise and slowly.

Changes made: Reasoning for the choice of dew point and slow addition of superacid is provided in page 5, lines 7-8 of the revised manuscript.

Reviewer comment 3. The authors baked the superacid treated TMD flake solutions. Did they wash the flakes before the next step in order to remove the superacid residues?

Author's response: The authors wish to thank the reviewer for this question. Yes, washing the superacid exfoliated flakes are absolutely essential prior to use. The flakes were washed with 1 liter of distilled water very cautiously inside the glovebox, and additional dilution was done with DI water so as to decrease the solution acidity.

Changes made: The washing process has also already been described, and is in lines 8-10, page 5 of the revised manuscript and has been also highlighted in yellow.

Reviewer comment 4. Please state in context the functions of PVDF, C black and NMP in the preparation of electrode?

Author's response: The author's wish to thank the reviewer for this question. C black is used as it helps to improve the electronic conductivity. PVDF is the binder and NMP is the solvent and is used to make the slurry in which PVDF, the active (electrode) material and C black is dissolved.

Changes made: The functionality of each of the components have been provided in the revised manuscript in page 6, line 18.

Reviewer comment 5. What are the sheet-edge distributed dots in the TEM image of MoTe₂?

Author's response: The author's wish to thank the reviewer for this very pertinent question. Per the authors, there are two possibilities for those dots. They are as follows:

Firstly, it can be assumed that the all of the bulk MoTe₂ obtained commercially may not have been in layered form and there may have been a very small fraction in a non-layered form. Consequently, on exfoliation, these non-layered parts did not exfoliate out and are observed as particulate "dots" in the TEM. Of course, they are a very small fraction of the entire sample and the TEM image proves that.

Secondly, it is also probable that those nanoparticles are some single-layered MoTe₂ that have crumpled/re-aggregated back together as this is a feature of exfoliated TMDs and a similar appearance has also been reported in the literature ^{1,2}.

Changes made: Information regarding the explanation of the appearance of these dots have been updated in the revised manuscript in page 9, lines 14-21.

Reviewer comment 6. In Figure 1, EDS data shows that there are relatively more C in the sample of MoS₂ than the other two. Why?

Author's response and changes made: The authors thank the reviewer for pointing it out and have uploaded fresh EDX data with no carbon signal in page 8 of the revised manuscript.

Reviewer comment 7. The author explained why in the TEM images there are no single-layered nanosheet. Is it also possible that after dilution, the exfoliated nanosheets tend to cluster due to Van der Waals interaction? This question is also somehow related to the purity of nanosheets (see question 3). For example, the Cl residue could exist in very small amount that the EDX data could not find it due to resolution issue.

Author's response: The authors wish to thank the reviewer for this question. The reviewer is correct; it is the tendency of exfoliated TMD nanosheets to cluster back due to the inherent van der Waal's forces acting at short distances ^{3,4}. That is why exfoliation is done using very strong chlorosulfonic superacid, which will help to prevent this restacking, and the authors have been able to employ this technique successfully in a previous report ⁵. Preliminary battery data also show reasonably good performance indicating restacking has not occurred immediately (20 cycles).

EDX (Fig. 1), Raman spectra (Fig. 2) and elemental mapping (Fig. 3) comprehensively demonstrates the formation of exfoliated layered TMDs, with no coexisting impurity phase present. However, it is also entirely true that the cleaning process, even though apparently showing no presence of Cl⁻ ion, there can be still some trace amounts left which is below the resolution limit of the EDX.

Changes made: Page 10, lines 2-3 in the revised manuscript has been updated about the trace amount of Cl⁻ residue.

Reviewer comment 8. Page 10 Line 38: there is a typo: "The peak positions and the peak position...".

Author response and changes made: The authors apologize for the error and have corrected the mistake, in line 18, page 10 of the revised manuscript.

Reviewer comment 9. Figure 2: please use arrows to help labeling the peaks. For 2(C), it looks like the sharp peak is the E_{2g} peak, which is different than the other two.

Response: The authors wish to thank the reviewer for this question. Per the suggestion of the reviewer, the peaks which were confusing have been designated with arrows, and the figure has been updated in the revised manuscript in page 10.

Also, the A_{1g} peak undergoes some broadening in MoTe₂ samples, which have been previously reported⁶. The sharpness of the peaks in the Raman depend upon which mode is Raman active, and in this case, it is observed that the in-plane vibrational mode i.e. the E_{2g} is significantly more Raman active⁷.

Changes made: The modified Raman spectra figure has been provided in the revised manuscript in page 10.

Reviewer comment 10. Figure 3: please use images with the same scale. Do all the prepared three sheet papers have the same thickness and internal "sandwich" cross section?

Author's response and changes made: The authors wish to thank the reviewer for this question. Yes, all the sheets have the comparable thicknesses and a similar “sandwich” like consistent cross-section.

All of the images in Figure 3 have a 10 um scale bar. From this, the reviewer can see that the thickness of the sheets are comparable with one other. As with any layered, packed structure of this nature, it is difficult/impossible to get them exactly the same.

Reviewer comment 11. Figure 3C: in term of the colors, it looks like the distribution of Mo and Te has a little local difference. Is that within the error range of EDX mapping?

Author's response: Yes, this local difference is within the error range of the EDX mapping.

Reviewer comment 12. Figure 4B and 4D: how large the error bar (repeatability)?

Author's response: The authors thank the reviewer for this question. The repeatability in this case was within $\pm 2\%$, experiments in triplicate were consistently agreeable.

Reviewer comment 13. Figure 4B and 4D: typo in the y-axis: “aerial” should be “areal”?

Author's response and changes made: The authors apologize for the error and thank the reviewer for pointing it out. The typo has been corrected in the updated figures 4B and 4D in page 13 of the revised manuscript.

Reviewer comment 14. Figure 5G: if TMD/rGO composite paper performs much better than pure rGO, why not preparing pure TMD paper?

Author's response: The authors wish to thank the reviewer for this question. The role of rGO is twofold: to significantly improve the electronic conductivity of bare TMDs and to enhance the structural integrity of bare TMDs as a composite.

Firstly, TMDs inherently have low electronic conductivities and so application of pure TMD papers will not yield practical results in a battery. Therefore, introduction of rGO is necessary to achieve appreciable conductivities⁸.

Secondly, the exfoliated TMDs tend to cluster/clump after a number of cycles due to the conversion reaction occurring and lose their sheet-like properties³. Therefore, rGO is used to make a composite as it acts as support for the TMDs and helps to relief stress during the intercalation process⁹.

Previous work by this same group have comprehensively demonstrated the advantage of using rGO as a composite along with exfoliated TMDs^{5, 10}.

Changes made: Information regarding the use of rGO have been added in page 17, lines 13-16 of the revised manuscript.

Reviewer comment 15. The author concluded that MoS₂ performs better in capacitor and battery than the counterparts, however no GCD and dQ/dV curves were shown for MoS₂.

Author's response: The author's thank the reviewer for this question. The figure is provided here for the reviewer's reference.

MoS₂ paper provides a first cycle desodiation (charge) capacity of 468.83.21 mAh g⁻¹. In figure (b), it is observed that certain peaks, especially at 0.8 V, 0.9 V and 1.74 V disappear after the first cycle, which indicate the formation of solid electrolyte interface layer. An insertion peak can be observed in the second cycle around 1.16 V, whereas the extraction peak is around 1.32 V. The trajectories of the galvanostatic charge-discharge curve and the dQ/dV curves are similar to the other TMD composite papers.

Changes made: The first and the second cycle GCD and dQ/dV curves for MoS₂, along with the explanation, have been updated in page 15 of the revised manuscript.

Reviewer 2.

Reviewer comment 1. As shown in Figure 1 (c), there are some nanoparticles on the edge of the layered MoTe₂. What are those particles?

Author's response: The authors wish to thank the reviewer for the question. Per the authors, there can be 2 possibilities of these nanoparticles that are observed. *Firstly*, it is possible that not all of the bulk MoTe₂ obtained commercially was in the layered form and a small fraction of it was in a non-layered bulk form, as a result the superacid based exfoliation process was unable to exfoliate out that non-layered part. This non-exfoliated part is observed as the nanoparticles.

Secondly, it may be also entirely possible that some of the exfoliated single layered MoTe₂ has restacked back and is observed in the form of discrete nanoparticles.

Changes made: Information regarding the explanation of the appearance of these dots have been updated in the revised manuscript in page 9, lines 14-21.

Reviewer comment 2. Stainless steel was used as the current collector for supercapacitor application, while a free-standing electrode configuration was adopted for sodium ion batteries. Why is that? Why is the free-standing electrode not used for supercapacitor characterization?

Author's response: The authors wish to thank the reviewer for this question. For battery application, a free-standing electrode was used as this arrangement facilitated optimum

performance, according to previous reports by the authors^{5, 10}. Also, for battery application, a compact CR2032 coin cell was used in which a free-standing electrode could be kept in place without it shifting and disrupting the proper functioning of the cell.

However, in the 3-electrode setup that has been used for supercapacitor application, it is necessary that a current collector is used, which also acts as a support. The set-up of the 3-electrode system is such that the electrode needs to be supported by a clip and a free standing electrode's structural integrity was compromised under the tensile stress from the clip. Hence, a stainless steel current collector was used during this project.

Also, we have determined experimentally that the stainless steel does not contribute to the capacitance in any appreciable way, moreover the supercapacitor performance was a qualitative determination of which TMD performed better and so the set-up does not really affect the fundamental result ($\text{MoS}_2 > \text{MoSe}_2 > \text{MoTe}_2$ in aqueous environment) of this manuscript.

Changes made: The reasoning behind the usage of stainless steel current collector have been added in page 7, lines 1-3 of the revised manuscript.

Reviewer comment 3. In Figure 4 (c), the charge-discharge curve for MoS_2 is not symmetric, suggesting a low coulombic efficiency. The authors should analyze and explain that. In addition, the electrochemical impedance spectroscopy measurements are suggested to be performed to better illustrate the differences among those TMDs.

Author's response: The authors wish to thank the reviewer for this question. It is true that the MoS_2 discharge curve in the 3-electrode setup is not symmetric, however, as pointed out before, the 3 electrode setup was just to qualitatively test the performance of these materials and confirm the results obtained with the battery (coin cell).

The impedance spectra was performed in 1M Na_2SO_4 electrolyte to maintain consistency and the Nyquist plot shows that for the three materials studied, the semi-circle, at the high-frequency region, is attributed to pseudocapacitive and double-layer (EDL) processes¹¹. Importantly, one fact that may contribute towards the fast-charge faradaic processes for TMDs is the fact that such species possesses different oxidation states¹². Thus, the capacitance may be a combination of EDL and faradaic charge storage processes. The equivalent circuit, per Tilak et al is provided below¹³.

From the circuit above, it is possible to obtain the contributions by each parameter i.e. bulk equivalent series resistance of the solution (R_{esr}), charge transfer resistance (R_{ct}), capacitance of the electric double layer ($C_{PE\ EDL}$) and the capacitance related to the pseudocapacitive reactions ($C_{PE\theta}$) and these values are provided in the table below.

Sample	R_{esr} (Ω)	R_{ct} (Ω)	$C_{PE\ EDL} - Y_0$		$C_{PE\theta} - Y_0$	
			n	$[\mu\Omega^{-1}\cdot s]^n$	n	$[\mu\Omega^{-1}\cdot s]^n$
MoS ₂	0.905	4.15	0.702	32.9	0.763	387
MoSe ₂	1.20	3.86	0.891	14.3	0.819	22.2
MoTe ₂	0.684	5.32	0.67	31.1	0.8	61.5

Changes made: Per the author's suggestion, EIS have been performed and is provided in the along with the detailed explanation, in pages 18 and 19 of the revised manuscript.

References:

1. Nicolosi, V.; Chhowalla, M.; Kanatzidis, M. G.; Strano, M. S.; Coleman, J. N., Liquid Exfoliation of Layered Materials. *Science* **2013**, 340.
2. Crane, M. J.; Lim, M. B.; Zhou, X.; Pauzauskie, P. J., Rapid synthesis of transition metal dichalcogenide-carbon aerogel composites for supercapacitor electrodes. *Microsystems and Nanoengineering* **2017**, 3.
3. Smith, R. J., Large-Scale Exfoliation of Inorganic Layered Compounds in Aqueous Surfactant Solutions. **2011**, 3944-3948.
4. Zhang, C., Polyphenol-Assisted Exfoliation of Transition Metal Dichalcogenides into Nanosheets as Photothermal Nanocarriers for Enhanced Antibiofilm Activity. *ACS Nano* **2018**, 12347-12356.
5. Romil Bhandavat, L. D., Gurpreet Singh, Synthesis of Surface-Functionalized WS₂ Nanosheets and Performance as Li-Ion Battery Anodes. *The Journal of Physical Chemistry Letters* **2012**, 1523-1530.
6. Vishwanath, S., MBE growth of 2H-MoTe₂ and 1T'-MoTe₂ on 3D. *Journal of Crystal Growth* **2018**, 61-69.
7. R. Saito, Y. T., S. Huang, X. Ling, M. S. Dresselhaus, Raman Spectroscopy of Transition Metal Dichalcogenides. *Journal of Physics: Condensed Matter* **2016**.

8. Schmidt, H.; Giustiniano, F.; Eda, G., Electronic transport properties of transition metal dichalcogenide field-effect devices: surface and interface effects. *Chemical Society Reviews* **2015**, *44*, 7715-7736.
9. Lee, J. E., Catalytic synergy effect of MoS₂/reduced graphene oxide hybrids for a highly efficient hydrogen evolution reaction. *RSC Advances* **2017**, 5480-5487.
10. Lamuel David, R. B., and Gurpreet Singh, MoS₂/Graphene Composite Paper for Sodium-Ion Battery Electrodes. *ACS Nano* **2014**, 1759-1770.
11. Nunes, W. G.; Vicentini, R.; Silva, L. M. D.; Costa, L. H.; Tadeu, T.; Zanin, H., Surface and Electrochemical Properties of Radially Oriented Multiwalled Carbon Nanotubes Grown on Stainless Steel Mesh. *Journal of the Electrochemical Society* **2018**, *165*, A3684-A3696.
12. Soon, J. M.; Loh, K. P., Electrochemical Double-Layer Capacitance of MoS₂ Nanowall Films. *Electrochemical and Solid State Letters* **2007**, *10*.
13. Tilak, B. V.; Chen, C. P.; Rangarajan, S. K., A model to characterize the impedance of electrochemical capacitors arising from reactions of the type. *Journal of Electroanalytical Chemistry* **1992**, *324*, 405-414.

Appendix B

June 21, 2019

RE: Submission of revised manuscript (**Manuscript id: RSOS 190437.R1**) titled, “*Exfoliated transition metal dichalcogenide (TMD) nanosheets for supercapacitor and sodium ion battery applications*”.

Dear Editor:

We are submitting the revised manuscript (**Manuscript id: RSOS 190437.R1**) titled, “*Exfoliated transition metal dichalcogenide (TMD) nanosheets for supercapacitor and sodium ion battery applications*”. Our submission will include the following:

1. This cover letter along with the following Appendix:
 - **Appendix I** which shows our response to the all the reviewer’s comments, attached with this cover letter.
2. **Revised primary manuscript file** (without any highlights and markups), which has all the updates and revisions we have made.

We have also used data for figures 6 (g), (h) and 7 (a) from a previous publication by our group (David et al., ACS Nano 2014, 8, 1759-1770), as a comparison and the copyright permission has been attached.

We would like to thank the editors for appreciating our manuscript and taking their time to provide valuable suggestions for further refining our manuscript. Look forward to hearing from you.

Sincerely,

Gurpreet Singh

Harold O. and Jane C. Massey Neff Professor,

Associate Professor, Mechanical and Nuclear Engineering Department

3002 Rathbone Hall, Kansas State University, Manhattan,

Kansas 66506, USA

<http://www-personal.ksu.edu/~gurpreet/>

Phone: 785-532-7085

Appendix I

Reviewer comments

Reviewer comment 1. It seems that the title for Figure 2 is not appropriate. There is no elemental analysis (mapping) shown in Figure 2. Instead, the elemental analysis is shown in Figure 3.

Author's response and changes made: The authors wish to thank the reviewer for this suggestion and Figure 2 (and 3) have been updated accordingly in the revised manuscript.

Reviewer comment 2. In Figure 4, the areal capacitance was calculated and plotted. I am wondering what the loading amount of the active material (TMDs) is for the supercapacitor test as the loading will affect the areal capacitance.

Author response: The effective mass loading on the electrodes for the supercapacitor measurements are usually 1 mg cm^{-2} and because of the coating uniformity, the variation of loading across samples is not more than $\pm 2 - 4\%$.

Reviewer comment 3. In Figure 5, the title for the x-axis in (b) should be as same as that in (d), (f) and (h).

Author response and changes made: The authors wish to thank the reviewer for this valuable suggestion and the figure (figure 6 in the revised manuscript) has been modified accordingly.

Reviewer comment 4. Table 1 should be reorganized.

Author response and changes made:

Reviewer comment 5: The part for electrochemical impedance spectroscopy is related to supercapacitor performance. I would suggest this part should be placed right after the supercapacitor performance.

Author response and changes made: The authors thank the reviewer for this valuable suggestion and the figure has been moved, and figure numbers have been updated in the revised manuscript.